# Bioactive Variability and In Vitro and In Vivo Antioxidant Activity of Unprocessed and Processed Flour of Nine Cultivars of Australian *lupin* Species: A Comprehensive Substantiation

**DOI:** 10.3390/antiox9040282

**Published:** 2020-03-27

**Authors:** Kishor Mazumder, Afia Nabila, Asma Aktar, Asgar Farahnaky

**Affiliations:** 1Department of Pharmacy, Jashore University of Science and Technology, Jashore 7408, Bangladesh; asmaaktar121039@gmail.com; 2School of Biomedical Sciences and Graham Centre for Agricultural Innovation, Charles Sturt University, Boorooma St, Wagga Wagga NSW 2127, Australia; asgar.farahnaky@rmit.edu.au; 3Department of Pharmacy, Faculty of Basic Medicine and Health Sciences, University of Science and Technology Chittagong, Foy’s Lake, Chittagong 4202, Bangladesh; nabila.afia726@gmail.com; 4School of Science, RMIT University, Bundoora West Campus, Plenty Road, Melbourne VIC 3083, Australia

**Keywords:** bioactive, oxidative stress, reactive oxygen species, lupin, functional food

## Abstract

The aim of this present investigation was to analyze bioactive compounds, as well as demonstrate the antioxidant activities of nine cultivars of Australian lupin species accompanied by observing the effect of domestic heat processing on their antioxidant activities adopting in vivo and in vitro approaches. Gas chromatography mass spectroscopy (GC-MS) analysis was performed for profiling bioactive compounds present in lupin cultivars. Multiple assay techniques involving quantification of polyphenolics, flavonoids and flavonol, electron transfer (ET) based assay, hydrogen atom transfer (HAT)-based assay and in vivo assays were performed. The major compounds found were hexadecanoic acid methyl ester, 9,12-octadecadienoic acid methyl ester, methyl stearate, lupanine,13-docosenamide and 11-octadecenoic acid (Z)- methyl ester. Mandelup was found to show excellent antioxidant activity. Moreover, Jurien, Gunyidi and Barlock had strong antioxidant activity. Both positive and negative impacts of heat processing were observed on antioxidant activity. Heating and usage of excess water during processing were the key determinants of loss of antioxidants. Negligible loss of antioxidant activity was observed in most of the assays whereas inhibition of both lipid peroxidation (33.53%) and hemolysis of erythrocytes (37.75%) were increased after processing. In addition, in vitro and in vivo antioxidant assays are found to show statistically significant (* *p* < 0.05 and ** *p* < 0.01) results, which are supported by the presence of a number of antioxidant compounds in GC-MS analysis.

## 1. Introduction

Oxidative stress triggered by excessive reactive oxygen species (ROS) may lead to cellular oxidative damage which is the root cause of life threatening pathological conditions such as diabetes mellitus, hypertension, obesity, cardiovascular diseases (e.g., atherosclerosis, myocardial infarction), carcinogenesis, neurodegenerative disorders (e.g., Alzheimer’s disease, Parkinson’s disease), cataracts, chronic renal failure, immune disorders, aging and others. Oxidative stress is the consequence of imbalanced redox homeostasis due to a lack of sufficient antioxidants that neutralize ROS. The key ROS, responsible for pathogenesis of these diseased conditions, include oxygen centered radicals such as superoxide (O_2_^−^•), hydroxyl radicals (OH•), alkoxyl radicals (RO•) and peroxyl radicals (ROO•), as well as oxygen centered non-radicals, such as hydrogen peroxide (H_2_O_2_), singlet oxygen (^1^O_2_), hypochlorous acid (HOCl) and ozone (O_3_) generated by several endogenous and exogenous mechanisms. These ROS are highly reactive due to having unpaired electron in atomic orbital and higher affinity to attack cellular components by either, donating own electron to or pulling out electron from them, thus, they can transfer their radical characters to the attacking species. The intake of exogenous antioxidant may be a promising approach to preventing or slowing down oxidative damage by neutralizing ROS, by interfering with its formation along with scavenging or decomposing it [1,2,3].

Phytochemicals having antioxidant properties are now in noticeable emphasis as the chronic diseases caused by oxidative stress become the key reason of mortality and inactivity. Plant-derived phenolic compounds (e.g., flavonoid, flavonol, flavanone, falvanonol and isoflavone) have strong antioxidant activities, which may suppress oxidative stress by combating ROS as well as maintaining redox homeostasis [4]. Antioxidants present in food are favorable over synthetic antioxidants because they contain multiple micronutrients in food that may act synergistically against chronic diseases caused by oxidative stress. Gas chromatography-mass spectrometry (GC-MS) has become steadfastly recognized as a key analytical platform for profiling secondary metabolites alike phenolics, steroids, alkaloids along with sugars, amino acids, fatty acids and others in plant, as well as non-plant sources [5,6,7,8]. The bioactive profile may help correlate the responsible key compounds required for distinct biological activities, as well as in revealing underlying mechanisms.

Lupin, a vital member of legume family pulls the attention of researchers because of its rich content of, not only antioxidants, such as phenolic compounds, flavonoids and tannins [9], but also essential amino acids, dietary minerals, higher proteins (~40%) and dietary fibres (~28%). Lupin was reported for having hypoglycemic, hypotensive, anti-atherosclerotic, antihyperlipidemic and gastroprotective action in both animal and human studies. Besides anti-carcinogenic, hepatoprotective, antiviral, antitumor, anti-HIV, antibacterial and antifungal effects are also reported. These pharmacological effects of lupin may be due to its antioxidant, anti-inflammatory properties, as well as special bioactive phytochemicals [10]. Lupin, having about 450 species can be grown in most parts of the world. Among them, Australian sweet lupin (*Lupinus angustifolius* and *Lupinus albus*) is the largest grain legume (500,000 to 1,000,000 tonnes annually) grown agro-economically in Western Australia due to favorable acidic sandy soils and Mediterranean climate [11].

It should be mentioned that phenolic content and antioxidant properties of lupin flour have been reported in a number of studies. A study reported that cooking process decreases antioxidant profiles of lupin significantly [12], whereas another studies suggested that cooking process may increase its antioxidant profile due to thermal breakage of cellular compartments and release of enclosed antioxidants [13]. As lupin flour is consumed mainly after processing or cooking, it is essential to investigate a comparative analysis of antioxidant properties between processed and unprocessed lupin flours.

In this present study an effort was made to quantify major antioxidant phytochemicals along with evaluating the antioxidant activities of unprocessed and processed flours of nine cultivars of Australian lupin seeds by using in-vitro and in-vivo approaches.

## 2. Materials and Methods

### 2.1. Chemicals

Isopropanol, methanol, ethanol, n-hexane, potassium persulfate (di-potassium peroxdisulfate), sodium acetate, chloroform and carbon tetrachloride (CCl_4_) were purchased from Merck (Darmstadt, Germany). Gallic acid, catechin, ascorbic acid, riboflavin, α,α-Diphenyl-β-picryl-hydrazyl (DPPH), 2,2′-Azino-bis (3-ethylbenzthiazoline-6-sulphonic acid) (ABTS), Folin–Ciocalteu (F–C) reagent and 6-hydroxy-2,5,7,8-tetramethylchroman-2 carboxylic acid (trolox) were obtained from Sigma–Aldrich (Steinheim, Germany). Ethylenediamine tetra-acetic acid (EDTA), nitroblue tetrazolium (NBT), dimethyl sulfoxide (DMSO), thiobarbituric acid (TBA) and trichloroacetic acid (TCA) were purchased from Sigma–Aldrich (St Luis, MO, USA). All the chemicals used for the study were high quality analytical grade.

### 2.2. Materials

Seeds of nine cultivars of two species of Australian lupin, *L. albus* cultivars: WK-338 (WK), Luxor (LUX), and Rosetta (ROS) and *L. angustifolius* cultivars: Jenabillup (JEN), Mandelup (MAN), Barlock (BAR), Jindulee (JIN), Gunyidi (GUN), and Jurien (JUR) were provided by Charles Sturt University, Australia.

#### 2.2.1. Preparation of Unprocessed and Processed Flour Samples

Lupin seeds were dehulled (Abrasion Debranner VTA5, Satake, Penrith Sydney, Australia) first to remove the seed coat. Then the seeds were ground using a Magic Bullet (Santos 01PV, Nella, Vaulx-en-Velin Cedex, Lyon, France) with same grinding time for each batch. Three different grinding batches were conducted for each cultivar. Half of flours for each of the cultivars were taken for processing in microwave (100 °C for 30 min) with excess water (flour: water = 1:3, *w*/*w*). Finally, the cooked flours were dried at 60 °C for 3 h to get processed flours.

#### 2.2.2. Extraction of Unprocessed and Processed Flours

A schematic diagram of entire methodology for extraction of processed and unprocessed lupin seed flours is shown in Figure 1.

### 2.3. GC-MS Analysis and Identification of Bioactive Compounds

GC-MS analysis of bioactive compounds present in unprocessed and processed lupin flours were accompanied by using a 7890A capillary gas chromatographic system (Agilent Technologies, Santa Clara, CA, USA) organized with a mass spectrometer and silica capillary column (HP-5MSI; length: 90 m, diameter: 0.25 mm, film: 0.25 µm) of 95% dimethyl-poly-siloxane and 5% phenyl.6 μL of the extract was run in a fused silica capillary column of with 99.999% helium (flow rate: 1 mL/min) as carrier gas. The injector temperature was 250 °C, ion-source temperature was 280 °C and isothermal temperature was 110 °C (2 min), with an increase of 10 °C/min to 200 °C then 5 °C/min to 280 °C and 9 min to 280 °C. The mass spectrum estimated was 50–550 m/z, where MS quad and source temperatures were maintained at 150 °C, and 250 °C, respectively. The overall GC-MS analysis time was 36 min. The spectrum of GC-MS was analyzed utilizing National Institute Standard and Technology (NIST) database [14] and the data was expressed as percentage of concentration.

### 2.4. Assessment of Antioxidant Activity

Antioxidant activity cannot be certified on the basis of a single antioxidant test model. A number of methods covering in-vitro and in-vivo techniques were reported by researchers. To get an overall estimation of antioxidant activity of processed and unprocessed extract of lupin flours of nine cultivars, the most commonly used methods were considered, including quantification of non-enzymatic antioxidant compounds, electron transfer (ET)-based and hydrogen atom transfer (HAT)-based in-vitro assay, anti-hemolytic activity assay, and biochemical assay of catalase (CAT), superoxide dismutase (SOD), lipid peroxidation (LPO) activities and protein content in serum, liver and kidney samples of experimental mice model. A summary of these adopted methods were shown in Table 1.

### 2.5. Assay of Non-Enzymatic Antioxidants

#### 2.5.1. Total Phenolic Content (TPC)

The TPC of processed and unprocessed lupin flour were determined by colorimetric Folin Ciocalteu method [15]. According to the assay, 250 µL of 10% (*v*/*v*) F–C reagent was added to 250 µL of sample (2 mg/mL) followed by vortexing and incubation (6 min, in dark). Subsequently a bluish color was developed after addition of 2.5 mL of 7% Na_2_CO_3_ followed by 90 min incubation at room temperature in the dark. Finally, absorbance was measured at 760 nm using a UV/visible spectrophotometer (Double Beam Spectrophotometer U-2900/2910, Hitachi, Minato-ku, Tokyo, Japan) and the values were read from gallic acid standard curve (*y* = 0.0081*x* − 0.0058; *R*^2^ = 0.9977). TPC was calculated according to the following equation as gallic acid equivalent (GAE) (mg/g):(1)$$Equivalent\;reagent\;\left(Conc.\right)\times \frac{Volume\;of\;total\;content}{Conc.of\;sample\;taken}.$$

#### 2.5.2. Total Flavonoid Content

A total of 0.5 mL of each sample extract solution (in methanol) was mixed with 0.3 mL NaNO_2_ (5%), the mixture was then incubated for 5 min at room temperature, followed by addition of 0.3 mL AlCl_3_ (10%). A yellowish color was developed by addition of 2 mL NaOH (1M). Then absorbance was measured by UV/visible spectrophotometer at 510 nm and the values were read from catechin standard curve (*y* = 0.0004*x* − 0.0058; *R*^2^ = 0.998). Total flavonoid content was calculated as catechin equivalent (CE) (mg/g) [16].

#### 2.5.3. Total Flavonol Content

To assess the total flavonol content of prepared extracts of processed and unprocessed flour of lupin, 2 mL of sample extract solution (0.1 mg/mL in ethanol) was mixed with 2 mL aqueous AlCl_3_ (20% *w*/*v*) with vigorously shaking and then incubation for 2.5 h at 20 °C. The total flavonol was analyzed by UV/visible spectrophotometer at 440 nm after development of a yellowish color and the values were read from a quercetin standard curve (*y* = 0.0381*x* − 0.0116; *R*^2^ = 0.9941). Total flavonol content was calculated as quercetin equivalent (QE) (mg/g) [17].

### 2.6. In-Vitro Evaluation of Antioxidant Activity

#### 2.6.1. Evaluation of Superoxide Radical Scavenging Activity

Superoxide radical scavenging activity was performed according to alkaline DMSO method [18]. 0.3 mL of each sample extract, the control (pure DMSO) and the standard (ascorbic acid) solution were prepared by dissolving into freshly distilled DMSO separately at various concentrations and transferred to reaction tube containing 0.1 mL of NBT (1 mg/mL in DMSO) followed by addition of 1 mL alkaline DMSO containing NaOH (5 mM) in 0.1 mL distilled water to get final volume of 1.4 mL. Then absorbance was measured spectrophotometrically using a UV/visible spectrophotometer at 560 nm and the % inhibition was calculated from the Equation (2):(2)$$\%\;Inhibition=\frac{Absorbance\;of\;control-Absorbance\;of\;standard\;or\;sample}{Absorbance\;of\;control}\times 100.$$

#### 2.6.2. Evaluation of DPPH• Radical Scavenging Activity

For evaluating DPPH radical scavenging activity of processed and unprocessed extract of lupin flours, reaction mixture was prepared by addition of 100 µL of sample extract and standard (ascorbic acid) at various concentrations into 2.5 mL of DPPH solution (0.3% methanolic solution) and 2.4 mL of methanol. These mixtures were allowed to stand 30 min at room temperature to complete the reaction. Pure DPPH in methanol was taken as positive control, whereas methanol was the blank. After changing the intensity of purple color to yellow, absorbance values were measured by a UV/visible spectrophotometer at 518 nm [18].

#### 2.6.3. Evaluation of Inhibition of LPO Using TBARS Method

The TBARS (Thiobarbituric Acid Reactive Substances) method was implemented to assess inhibition of LPO of processed and unprocessed extract of lupin flour by using egg homogenate [19] and bovine brain homogenate [20]. Egg homogenate (10% *v*/*v* in distilled water) was prepared by homogenization and 0.50 mL of it was added to the various concentrations of sample extract solution (in phosphate buffer) and standard (ascorbic acid). In order to induce lipid peroxidation reaction, 0.50 mL FeSO_4_ (0.07M) was added and subsequently incubated for 30 min. When the colour changed to red, the reaction was terminated by the addition of 1.5 mL of 20% acetic acid (pH 3.5), 1.5 mL of 0.8% TBA and 0.05 mL of 20% TCA. Then, the mixture was vortexed, heated (60 min), and cooled, respectively, and followed by the addition of 1-butanol (5.0 mL) and centrifugation (3000 rpm, 10 min). Finally the absorbance of the supernatant was measured at 532 nm.

#### 2.6.4. Evaluation of Hydroxyl Radical Scavenging Activity

To assess hydroxyl radical scavenging activity deoxyribose method was used [21]. Reaction mixture was prepared by addition of 0.2 mL of each reagent including 3mM deoxyribose, 0.1 mM ferric chloride, 0.1 mM EDTA, 0.1 mM ascorbic acid and 2 mM hydrogen peroxide (in 20 mM phosphate buffer, pH 7.4). 0.2 mL of sample extract solution (in DMSO) at various concentrations was added to reaction mixture to have final volume of 1.2 mL. After incubation (30 min, 37 °C), ice-cold TCA (0.2 mL, 15% *w*/*v*) and TBA (0.2 mL, 1% *w*/*v*) contained in 0.25N HCl were added and allowed to keep on water bath (30 min) for inducing color change (yellow to purple). After that, the mixture was cooled, and absorbance was measured at 532 nm. The percentage inhibition of hydroxyl radical scavenging activity was calculated from the Equation (2).

#### 2.6.5. Evaluation of ABTS^+^• Radical Scavenging Activity

ABTS^+^**•** radicals were generated from the reaction of a strong oxidizing agent, 2.54 mM potassium persulfate (K_2_S_2_O_8_) with 7 mM solution of ABTS salt. The radical solution was made ready for use by standing 15 h in dark at room temperature. Reaction mixture was prepared by addition of 0.5 mL of sample extract at various concentrations to 0.3 mL of ABTS^+^**•** radical solution followed by adjustment of final volume to 1 mL with ethanol. After discoloration (purple to colorless), absorbance was measured immediately at 734 nm by a UV/visible spectrophotometer. The results were expressed compared with trolox (water soluble vitamin E analogue) standard [22].

#### 2.6.6. Evaluation of Hydrogen Peroxide (H_2_O_2_) Scavenging Activity

H_2_O_2_ scavenging activity of processed and unprocessed extract of lupin flour was assessed by spectrophotometric measurement of the comparative loss of H_2_O_2_ after scavenging by sample extract and ascorbic acid (standard). For performing the assay, phosphate-buffered saline (pH 7.4, 50 mM) was used to prepare 40 mM H_2_O_2_ solution. 2 mL of the H_2_O_2_ solution was mixed with 1 mL of sample extract at various concentrations followed by 10 min incubation. Then absorbance was measured by a UV/visible spectrophotometer at 230 nm [23].

#### 2.6.7. Evaluation of Anti-Hemolytic Activity

Anti-hemolytic activity of the extract was evaluated by using the method of Ebrahimzadeh et al. [24] with slight modification for human red blood cells (RBCs). Human blood was centrifuged to separate human RBCs and washed with phosphate buffered saline (PBS) (pH 7.4). Diluted suspension of RBCs (4% in PBS) was prepared and 2 mL was added with 1 mL of sample extract solution (in PBS) at various concentrations followed by adjusting final volume to 5 mL with PBS. The mixture was then incubated for 20 min at room temperature. Oxidation of membrane phospholipid of RBCs was initiated by addition of 0.5 mL of H_2_O_2_ solution (in PBS), then shaking for 40 min at 37 °C. The final mixture was centrifuged (1500 rpm, 10 min) and the extent of hemolysis was assessed by using a UV/visible spectrophotometer at 540 nm.

### 2.7. In-Vivo Assay of Antioxidant Activity

Amongst nine cultivars of Australian lupin, Mandelup showed the most potent antioxidant activity in in-vitro studies. Various types of in-vivo methods mentioned (Table 1) were further executed with the aim of confirming antioxidant potential of unprocessed and processed flours of Mandelup in relevant biological environments.

#### 2.7.1. Experimental Animals

In order to conduct the experiments, Swiss albino mice of either sex having 25–30 g body weight (bwt) were used which were purchased from animal research branch of icddr,b, Dhaka, Bangladesh. They were kept in standard environmental conditions (22 ± 2 °C, 55 ± 5% R.H., 12 h light and 12 h dark cycle) for 1 week before experiments to accustom and fed with commercial pellet diet and water *ad libitum*. Protocols (Ref: ERC/FBS/JUST/2018-38, approved on 16 February 2018) adopted in the studies were approved by institutional animal ethical committee of Jashore University of Science and Technology, Jashore.

#### 2.7.2. Experimental Design

In-vivo studies were performed as per CCl_4_ induced oxidative toxicity method [28]. A total of 24 mice of either sex were randomly divided into seven groups where each group contained six mice. Group I (control) had free access to normal diet. Group II (CCl_4_ control) received CCl_4_ in olive oil (1:1) at a dose of 2 mL/kg bwt by intraperitoneally (IP) on 1st and 7th day of experiment. After 48 h of CCl_4_ treatment, group III (standard) was given silymarin (25 mg/kg bwt) and group IV, V, VI and VII (test) were given unprocessed and processed flour extract at the dose of 200 mg/kg and 400 mg/kg bwt respectively. Standard and flour extracts were administered by IP route. After 24 h of the end of treatment serum were collected after anesthetizing in ether chamber and then the animals were sacrificed by cervical dislocation. Liver and kidney samples were removed and cleaned with ice-cold saline (0.9% NaCl) for biochemical investigation.

#### 2.7.3. Biochemical Assay of CAT, SOD, MDA Activity and Protein Content

Liver and kidney homogenates equal to 10% were prepared by using ice-cold potassium phosphate buffer (pH 7.4). After centrifugation (1000 rpm, 10 min) of the homogenates and serum sample, supernatants were collected and stored at 4 °C for biochemical assay of antioxidant activity.

#### 2.7.4. Assay of CAT Activity

Activity of CAT was assessed by the method of Sinha et al. [25] in liver, kidney and serum sample. 50 μL of supernatant was added to the reaction mixture consisting of 2 mL of phosphate buffer (pH 7.4) and 1 mL of H_2_O_2_ solution (30 mM) contained in a cuvette. After incubation (30 min, 37 °C), CAT activity was measured at 240 nm.

#### 2.7.5. Assay of LPO Activity by TBARS Method

The TBARS assay reported by Ohkawa et al. [19] was used to determine malondialdehyde (MDA) content (form of TBARS) generated from peroxidation of lipid in liver and serum samples. According to this assay, the supernatant (50 μL) was mixed with TCA (14%) and TBA (0.6%) for de-proteinization. Then the reaction mixture was heated (30 min), followed by cooking (5 min). After centrifugation (2000 rpm, 10 min) of the mixture, LPO activity was measured at 535 nm.

#### 2.7.6. Assay of SOD Activity

SOD activity was assayed in liver and serum sample by observing the inhibition of auto-oxidation of riboflavin as per the method of Mccord et al. [26]. Chloroform (0.25 mL) and ethanol (0.5 mL) were mixed with 1 mL of supernatant and centrifuged at 1800 rpm for 6 min. 100 μL of supernatant was diluted with phosphate buffer (pH 7.4) to adjust volume of 2.25 mL and transferred to reaction tube containing EDTA (0.2 mL), NBT (0.1 mL) and riboflavin (0.5 mL). SOD activity was measured at 560 nm after changing the intensity of colour.

#### 2.7.7. Estimation of Protein Content

The protein content of the liver sample was determined by the method of Lowry et al. [27] and 0.2 mL of supernatant was added to Lowry solution and incubated at room temperature for 10 min in dark condition. Then, F–C reagent (0.2 mL) was added to the mixture followed by incubation for 30 min. Protein content was estimated at 750 nm by a UV-visible spectrophotometer and the corresponding values were read from bovine serum albumin (BSA) standard curve.

### 2.8. Statistical Analysis

Statistical analysis was accomplished with two-way ANOVA followed by Duncan’s multiple range test designed for making comparison amongst experimental groups using SPSS 23.0 software (IBM, New York, NY, USA). Student’s t test was performed to make comparison between unprocessed and processed condition. The results were considered significant at differences of * *p* < 0.05 and represented as mean ± standard error of mean (SEM).

## 3. Results and Discussion

### 3.1. GC-MS Analysis and Identification of Bioactive Compounds

GC-MS analysis was performed for bioactive compounds profiling of all nine varieties of *L. angustifolius* and *L. albus* seed flours in both unprocessed and processed conditions. A number of bioactive compounds having biochemical and structural differences are found to be present (Appendix A and Figure 2). An example of TIC for unprocessed and processed flours of JEN cultivar was shown in Appendix A. Six common compounds, namely hexadecanoic acid methyl ester, 9, 12-octadecadienoic acid methyl ester, methyl stearate, lupanine, 13-docosenamide and 11-octadecenoic acid (Z)- methyl ester present in both, unprocessed and processed flours of JEN. Few compounds (such as methyl tetradecanoate) disappeared after processing whereas a number of new compounds (such as malic acid, anethole and heterocyclic compounds) were found in processed condition. Thermal decomposition may reduce or eliminate any compounds whereas release of unbound chemicals from cell compartment in the presence of heat may cause appearance of new compounds, as well as increase of existing compounds.

All the cultivars were found to enclose diverse bioactive compounds having numerous biological activities in GC-MS data (Appendix A). The results of GC-MS analysis revealed the uniqueness of six major compounds namely hexadecanoic acid methyl ester, 9,12-octadecadienoic acid methyl ester, methyl stearate, lupanine,13-docosenamide and 11-octadecenoic acid (Z)- methyl ester present in both unprocessed and processed flours of all nine cultivars (Figure 2). Two major compounds: pyrrole-(1,2,a)-pyrazine-1,4-dione, hexahydro-3-(2-methyl)propyl and 4H-pyran-4-one-2,3 dihydro-3,5-dihydroxy were present in Jenabillup (unprocessed and processed), seven major compounds: 2-pyrrolidione, 2(3H)-furanone, dihydro-5-(2-octenyl)-(Z), gamma-dodecalactone, 2 piperidinone, 4-methylenebicyclo(4.2.0)oct-2-ene, dodecanamide, N-isobutyl, 4 nitrobenzoylmethyl-.beta.-phenylpropionat and pyrrolo(1,2-a)pyrazine-1,4-dione were found in Jindule (unprocessed and processed), three major compounds: squalene, hexadecanoic acid, 2-hydroxy-1-(hydroxymet.) and di-n-octylphthalate were present in both unprocessed and processed Jurien, one major compounds: 13-hydroxy-lupanine TMS derivative was present in WK (unprocessed and processed) (Appendix A). Lupin cultivars were rich in quinolizidine alkaloid (QA) and fatty acid methyl ester. QA like lupanine, 13-OH-lupanine and 13a-acetoxylupanine were found in various cultivars. Biosynthesis of QA alkaloids in lupin cultivars were mediated by genetic factors as well as abiotic stress [29]. 14 different fatty acid methyl esters were present. Moreover, a volatile oil (anethole), a phytosterol (campesterol), a chromo-peptide (actinomycin C2), amino compounds, numerous heterocyclic compounds were found in various cultivars. A number of phytochemicals disappeared in the processed condition as compared to the unprocessed condition. Contrariwise, several phytochemicals were increased in most of the cultivars after processing. These changes in phytochemical constituents could be due to thermal decomposition and/or chemical modification of phytochemicals during boiling [30,31]. Volatile compounds like anethole were more prone to loss after processing. Most of those QA, fatty acid methyl ester, amine and heterocyclic compounds were reported for antioxidant activity. Antimicrobial, antiprotozoal, hypocholesterolemic, anti-cholinesterase, analgesic, gastro-protective and anti-diabetic activities were also reported for many compounds. Several potent cytotoxic bioactive compounds were also found which were reported for potential anticancer activity, alike pyrrole (1, 2, a) pyrazine 1, 4, dione, hexahydro 3-(2-methyl propyl) [32], 7,10-hexadecadienoic acid, methyl ester [33], and 1-monolinoleoylglycerol trimethylsilyl ether [34]. Furthermore, compounds having antimicrobial potential were found in lupin flours, e.g., actinomycin C2, 13a-acetoxylupanine, methyl tetradecanoate, 2,4-decadienal, (E,E)-, and anethole [35,36,37,38,39].

Six major compounds were present in the nine cultivars with varying concentrations which were mostly declined except lupanine and 13-docosenamide after heat processing (Table 2). Most of the common six compounds were fatty acid methyl esters except lupanine and 13-docosenamide. Lupanine is a well-known quinolizidine alkaloid and 13-docosenamide is an amino compound. Increase of lupanine and 13-docosenamide after processing in all cultivars was dependent on a number of factors, such as higher boiling points made them non-volatiles, stable at processing temperature (100 °C) and release of unbound form in presence of heat. The four fatty acid methyl esters are volatile and are susceptible to loss in the presence of heat. The variability of the concentration of these bioactive compounds among cultivars might be due to the exposure of plant to environmental temperature, light, stressed conditions, as well as genetic factors [40]. These six common compounds were reported to have several biological activities by many researchers. Lupanine was reported to possess antioxidant (inhibition of lipid peroxidation), anti-cholinergic as well as antidiabetic activity by improving glucose homeostasis along with stimulating insulin secretion [41,42,43,44,45]. Researchers reported n-Hexadecanoic acid to have antioxidant, anti-inflammatory, hypocholesterolemic and cancer prevention activities [33], 9,12-octadecandionoic acid to have anti-inflammatory, antibacterial, hypocholesterolemic and hepatoprotective activities, 11-octadecenoic acid, methyl ester to have antioxidant and antimicrobial properties [46], methyl stearate to have anti-inflammatory, intestinal lipid metabolism regulation, nematicidal, antinociceptive, antioxidant and antifungal activities and 13-docosenamide to have antimicrobial activity [47,48].

### 3.2. Assay of Non-Enzymatic Antioxidants

#### 3.2.1. Total Phenolic, Flavonoid and Flavonol Content in Lupin Flours

Phenolics comprise a large group of vital antioxidant family present in Australian sweet lupin seeds which exists as phenolic acid (caffeic, *p*-coumaric, ferulic, rosmarincic, chlorogenic, vanillic, protocatechuic and *p*-hydroxybenzoic acid mainly), flavonoids (aglycone of luteolin, apigenin and diosmetin), flavonols (quercetin, rutin, kaempferol, myricetin and others), isoflavones (genistein derivatives, phytosterols) and flavones [10,49,50]. Total phenolics, flavonoids and flavonols of nine cultivars of lupin seed flours were assayed in this current study and observed results areshown in Figure 3. Among the nine cultivars, Mandelup (MAN), Gunyidi (GUN) and Barlock (BAR) which are the members of *L. angustifolius* species showed the most significant TPC in unprocessed state (17.11 ± 0.93, 18.10 ± 2.01 and 20.69 ± 1.09 mg/g respectively) whereas WK-338 (WK), a *L. albus* species showed the least (9.70 ± 0.89 mg/g) TPC. The total flavonoids and flavonols present in the nine cultivars followed the result observed from TPC assay. MAN, GUN and BAR were found to be rich in flavonoids (55.32 ± 1.88, 49.93 ± 0.91 and 46.92 ± 2.01 mg/g) and flavonols (30.14 ± 0.79, 17.64 ± 1.01 and 19.58 ± 1.19 mg/g) also. WK was found to contain least amount in both cases like phenolics. Both the TPC (9–20 mg/g) and flavonols (13–30 mg/g) were satisfactory in all cultivars, but flavonoid (27–55 mg/g) content was highly significant. All the cultivars contained >30 mg/g flavonoids except WK. Cultivars of *L. angustifolius* species were found to be more potential sources of antioxidants compared to the cultivars of *L. albus* species. Siger et al. studied three species of lupin and revealed that *L. angustifolius* has better antioxidant profiles, specially phenolics and flavonoids than that of *L. albus* which matched the obtained results of current study [9]. Kalogeropoulos et al. showed a comparative analysis of total phenolics in different types of legumes, such as lupin, black-eyed peas, broad beans, chick peas, lentils, pinto beans, split peas and white beans, in which lupin had found to contained about 19.4 mg/g phenolics and highest content of phytosterol (one type of flavonoid) (53.6 mg/g). Other legumes contained significant amount of polyphenolics (12.7–25.85 mg/g) but phytosterol content were very poor [12]. Oomah et al. investigated the total phenolics and flavonoids content of eight genotypes of *L. angustifolius* and concluded that biological antioxidant activities of phenolic group are less affected by genotypes of lupin [51].

Generally, phenolics are ubiquitously present in plants and synthesized in response to microbial infections. Among all phenolic groups, flavonoids perform as secondary defense compounds in several plants, by presenting in the ROS generation region of mesophyll cell nucleus and synthesized in excess amount [52]. Variances of phenolic, flavonoid and flavonol antioxidants among lupin cultivars might be the result of diversity in agro-environmental conditions, biochemical differences in metabolic processes, disease state and defense mechanism [10]. Mutations and cultivation conditions (fertilizers, insecticides, watering, sun-ray and so others) might add the possibility of these variations. Adopted assay techniques might cause over/under-estimation of the compounds. For example, Folin Ciocalteu method has poor selectivity for phenolics. Other compounds present in samples (such as nitrates, carbohydrates) can react to F–C reagent which leads to over-estimation. Moreover, anti-nutrients, such as alkaloids present in lupin might interact with reagents and cause interference in estimation of targeted phenolics [9].

#### 3.2.2. Effect of Processing on Total Phenolic, Flavonoid and Flavonol Content

Processing showed positive impact on total phenolics but negative impact on total flavonoids and flavonols. An increase of 1.5–14 mg/g phenolics was observed in lupin seed flours after processing. A significant increase (>10 mg/g) in TPC was found mostly in *L. albus* species. TPC in MAN and GUN cultivars were also increased satisfactory. The positive change of TPC due to domestic processing was also supported by other authors [13,53,54,55]. The findings from some other studies [12] were contradictory which stated that heating and boiling caused significant decrease of phenolics. The possible explanation of increased TPC after processing might be leaching bound phenolic constituents out of the cellular organelles of lupin seeds. Inactivation of the polyphenol oxidase enzyme during heating may reduce the thermal oxidative degradation of polyphenolics present in lupin seeds.

A diminution of total flavonoids (0.9–18.9 mg/g) and flavonols (1.83–7.07 mg/g) were found after processing. Reduction of flavonols was not momentous where drastic loss of flavonoids (>10 mg/g) was found in all cultivars except Jenabillup. Yu et al. [55] found the same fate of flavonoids and flavonols after heating in case of purple skin eggplants. Most of the researchers agreed with the findings. Some contradictory findings were also found which noted positive effects of heating on flavonoids and flavonols. Basically, flavonoid and flavonols contain hydroxyl group as functional moiety of antioxidant action. Its chemical stability depends on structure, number of hydroxyl groups and magnitude of polymerization, substitution and conjugation [52]. All these variables are heat sensitive. Synergism or antagonism of biochemical reactions accelerated or declined by heat may decrease the bioactive antioxidants or increase strong oxidants. Moreover heat sensitive flavonoids and flavonols might undergo thermal degradation. In addition, a number of hydroxyl groups in flavonoids make them water soluble. During processing, they are prone to loss with excess water used for boiling.

A remarkable upshot was noted by Chaaban et al. [56] after studying degradation kinetics of flavonoids. He stated that degradation of flavonoids initiates at lower temperature (<35 °C). Rutin and quercetin derivatives (flavonols mainly) are less affected by temperature below 100 °C Antioxidant activity of these flavonoids and flavonols depend solely on the nature of degradation products which might have superior, similar or inferior antioxidant activity than the native compounds. Thermal glycosylation and methylation reduce antioxidant activity whereas dimerization increases it. These statements clarify the possible causes of positive change of TPC along with negative impact on flavonoids and flavonols. Degradation products might loss flavonoid and flavonol character but retain phenolic nature as polyphenolics include a vast class of compounds. For example, degradation of rutin forms procatechuic acid which is not a flavonoid or flavonol but a potent phenolic acid.

### 3.3. In-Vitro Evaluation of Antioxidant Activity

#### 3.3.1. Evaluation of Superoxide Radical Scavenging Activity

Superoxide (O_2_^−^•) is a physiologically important anion, which can be dangerous in excessive amounts. Its overproduction may lead to severe damage of proteins, DNA and mitochondria [3]. In the present study, superoxide radical scavenging activities of unprocessed and processed flours of lupin were assayed and the observed results were expressed in Figure 4. All the cultivars showed excellent inhibition (79.47 ± 2.12% to 96.22 ± 1.56%) compared to ascorbic acid standard (98.98 ± 3.11%) at 2 mg/mL concentration. The inhibition followed dose-dependent manner. IC_50_ values (Table 3) of the most effective cultivars MAN, JUR, BAR and GUN were 552.2, 535.8, 376.2 and 366.2 μg/mL respectively compared to ascorbic acid which was 276 μg/mL. After processing, the inhibition was diminished from 1.94 to 24%. IC_50_ values were increased to 604.9, 679.8, 759.0 and 465.1 μg/mL for MAN, JUR, BAR, and GUN, respectively. The presence of flavonoids and flavonols in lupin cultivars is one of the key reasons for higher superoxide radical scavenging activity. Heat processing, along with boiling with excess water, reduced the flavonoids and flavonols content that affected superoxide radical scavenging activity. A number of studies found the same fate of flavonoids and flavonols after heat processing and boiling with water in natural foods such as vegetables, spices and legumes [13,55,57,58]. In addition, GC-MS data revealed that most of the major bioactive compounds either, decreased or disappeared after processing. From the obtained results and observations of previous studies, it can be concluded that flavonoids and flavonols being washed out with water was more effective than the heating process.

#### 3.3.2. Evaluation of DPPH• Radical Scavenging Activity

DPPH• radical scavenging assay is the most widely adopted antioxidant assay technique used for natural drugs and foods due to its easy and fast nature along with reliable outcome [3]. It is a hydrophobic radical which may react with less polar antioxidants transferring capacity [59]. The DPPH• Radical Scavenging activity of processed flour was decreased due to cooking and gradually increased according to concentration (Figure 5). In this study, Jindulee showed better scavenging at 2 mg/mL conc. (90.736 ± 2.56%) compared to standard ascorbic acid (96.319 ± 2.039%) which was significantly reduced to 65.82 ± 2.86% after processing. IC_50_ values for highly active cultivars MAN, JUR, BAR and GUN were 193.8, 676.6, 591.9 and 660.9 μg/mL respectively compared to standard ascorbic acid, 56.6 μg/mL. After processing, they were reduced significantly. No correlation was found between DPPH• scavenging activity and TPC in this study. Active antioxidants and antioxidant activities are completely different as there are several factors which affect the antioxidant activities, such as: (1) formation of less active compounds due to heating, (2) formation of pro-oxidants, (3) structural relationship with activities (4) affinity and binding ability of reagents with antioxidants present in a sample [59,60,61,62]. In addition, active antioxidants were also affected by heat processing. According to GC-MS data, pyrazine, trimethyl and tetramethyl; 1,2,3-butanetriol; pyrazine, 3-ethyl-2,5-dimethyl- and 2-methyl-4-vinylphenol were declined after processing which were reported for free radical scavenging property [63,64,65].

#### 3.3.3. Evaluation of Hydrogen Peroxide, Hydroxyl and ABTS^+^• Radical Scavenging Activity

Hydroxyl radical (OH•) is foremost harmful ROS in biological systems which attacks DNA, RNA, proteins and even cell membranes. Hydrogen peroxide (H_2_O_2_) is a weak oxidant responsible for inactivation of thiol group (–SH) containing enzymes and ROS generation by OH• formation within cells in reaction with divalent cations (such as Fe^2+^, Cu^2+^). ABTS^+^•, a strong hydrophilic radical can be neutralized by hydrogen atom and this assay is used to examine antioxidant activities of a mixture of compounds [3,59]. OH•, H_2_O_2_ and ABTS^+^• assays were performed in this investigation to assess the overall estimation of hydrogen atom donating (HAT) abilities as well as synergistic or antagonistic effects of lupin flours. The obtained findings of these assays were shown in Figure 6 and Table 3. MAN was found to show the best activities in these assays. Among these three assays, lupin flours were found to be mostly active against OH• radicals (82.07 ± 4.23 to 95.896 ± 4.34% inhibition) and ABTS^+^• radicals (85.70 to 90.68% inhibition). Inhibition activities against H_2_O_2_ were moderate (63.24 ± 3.31 to 85.29 ± 3.93% inhibition). After domestic processing, OH• inhibition activity was affected severely (1.21–26.44%) whereas the changes in case of H_2_O_2_ and ABTS^+^• radical inhibition activities were not substantial. IC_50_ values for ABTS^+^• radical (366, 156.2 and 349.6 μg/mL for MAN, BAR and GUN respectively) were significant compared to standard trolox (137μg/mL). From these evidences, it can be concluded that lupin cultivars possess hydrogen atom donating and electron transferring capacity and can stabilize hydrogen peroxide, thus inhibit formation of reactive OH• and other free radicals along with scavenging them.

Lupin cultivars contain a number of phenolic acids, flavonoids and flavonols. In addition, they are good sources of protein, especially histidine containing proteins. Antioxidants along with histidine containing proteins lead to synergistic effects of lupin flours in these assays, exclusively in ABTS assay [10,59]. GC-MS data indicated that lupin is a good source of 1,2,3-butanetriol and pyrazine derivatives having proven free radical scavenging, as well as H_2_O_2_ stabilization property [63,64,65]. Divalent cations present in cells act as pro-oxidant and initiate H_2_O_2_ breakdown along with ROS generation. Alkaloids are a potential chelator for binding metallic ions [66]. Lupin cultivars are good sources of alkaloids, which add fuel in H_2_O_2_ stabilizing as well as free radical scavenging capacity.

#### 3.3.4. Evaluation of Inhibition of LPO Activity Using TBARS Method

TBARS assay is widely used by researchers to establish antioxidant activity against lipid peroxidation of unsaturated lipid. Here, TBARS assay was performed using egg-yolk homogenate and bovine brain homogenate as lipid sources and the results were displayed in Table 4. It was found that the inhibition of LPO activity in egg-yolk homogenate of lupin flour was increased considerably after processing as well as with concentration. Jindulee (JIN) showed minimal LPO activity (61.01 ± 2.26%) compared to standard ascorbic acid (92.89 ± 3.03%). LPO for brain homogenate adjoined the same finding. Jurien showed better inhibition at 2 mg/mL conc. for the unprocessed flour (88.249 ± 3.34%). MAN, JUR, BAR and GUN showed the most significant LPO inhibitions. IC_50_ values for these cultivars were 487.7, 323.9, 530.8 and 584 μg/mL respectively, compared to ascorbic acid (91.3 μg/mL).

Significant amplification of LPO activity was observed for JIN (95.534 ± 3.06%) after processing. MAN, BAR and JUR showed significant increases in LPO inhibition after processing. IC_50_ values for these cultivars were augmented to 369.8, 157.7 and 368 μg/mL compared to unprocessed flour. Bovine brain homogenate also supported the findings.

There was a correlation between TPC and LPO activity. Thermal processing increases TPC that accelerates LPO inhibitions. Specially, phenolic acids (gallic acid) have strong LPO inhibition activities which are reported by a number of studies [67,68,69,70]. Moreover epicatechin and rutin present in lupin have strong LPO inhibition activity [71]. In the presence of heat, rutin undergoes dimerization and other chemical reactions leading to form dimer of rutin, procatechuic acid and stable orthosemiquinone radicals having stronger LPO inhibitory activities [60,72]. The lupin species was found to contain lupanine in GC-MS analysis, which was increased significantly after processing. Lupanine is a quinolizidine alkaloid having excellent LPO inhibition potential [44,45]. These are the possible reasons behind the increase of LPO inhibitory activity of lupin flours after processing. This result supports lupin as a potential source of antioxidants able to defend against LPO of biological membranes and others unsaturated lipid tissue of human body.

#### 3.3.5. Evaluation of Anti-Hemolytic Activity

Membrane phospholipids of erythrocytes undergo lipid peroxidation in the presence of H_2_O_2_ which results in membrane degradation and hemolysis leading to dangerous cell viability and progression of various types of cancers [72]. Substances having long term antioxidant property, H_2_O_2_ stabilizing capability and LPO inhibiting activity may resolve the damage and stabilize erythrocytes. In this study, the anti-hemolytic activity of lupin flours was determined and the results were shown in Figure 7 and Table 3. Anti-hemolytic activity of lupin flour was increased after processing as well as concentrations. Most of the cultivars possess marginal anti-hemolytic activity which was 62.5 to 80.6% and significant increase of this activity (8.60–37.75%) was observed after processing. IC_50_ values for most effective cultivars MAN, JUR, BAR and GUN were 426.2, 475.5, 580.2 and 572.1 μg/mL correspondingly after processing compared to standard ascorbic acid (237.1 μg/mL). The mechanism behind this high anti-hemolytic activity of lupin might be additive or synergistic effects of numerous types of antioxidants having multiple functionalities. Such as, (1) direct inhibitory effects on membrane LPO of several phenolic acids including ferulic acid and chlorogenic acid [73], (2) thermal acceleration of H_2_O_2_ stabilizing and LPO inhibiting activities of rutin [52], (3) iron and copper ion chelation activity of quercetin [66] and others. Moreover, thermal processing makes the bioactive compounds (such as lupanine) more accessible by extracting them from cellular matrix and several other mechanisms described above. These statements prove that lupin cultivars possess potential anti-hemolytic activities.

### 3.4. In Vivo Assay of CAT, SOD, LPO Activity and Protein Content

The CAT, SOD, LPO activity and protein content assay were performed to ensure antioxidant activity of Mandelup which was observed as the most effective antioxidant containing cultivar of lupin in overall in vitro assay. CCl_4_ treatment in mice model induces oxidative damage by a number of mechanisms involving acceleration of ROS generation, reduction of SOD and CAT activity, induction of LPO and interrupting others oxidative defenses. Metabolism of CCl_4_ by cytochrome P450 monooxygenase enzyme system leads to formation of trichloro-methyl (CCl_3_•) and trichloro-methyl-peroxyl (•OOCCl_3_) radicals. These highly toxic and reactive radicals initiate chain oxidation reactions by binding to cellular proteins, membrane phospholipids and unsaturated fats covalently. They also interrupt the cellular enzymatic antioxidant defense system [74,75,76].

CAT and SOD are considered as first line antioxidant defense enzymes. CAT metabolizes reactive and unstable H_2_O_2_ to cellular oxygen molecules and water [77]. SOD is the scavenger of ROS along with modifying ROS and unstable free radicals to comparatively stable compounds. For example, SOD coverts ROS to less reactive H_2_O_2_ [78]. MDA is the key metabolite of LPO which induces oxidative damage resulting in severe pathological conditions [79].

The in vivo activities performed in various biological samples collected from experimental mice models were shown in Figure 8. In this study, CAT and SOD activity of Mandelup followed the order: liver > kidney > serum and liver > kidney respectively. The activities were concentration dependent in both cases. Effect of processing was minor in CAT activity whereas significant loss of SOD activity was observed. At 400 mg/kg dose, CAT activity in liver, kidney and serum were 77.26, 56.68 and 37.84% respectively, which were reduced <10% after processing. SOD activity in liver and kidney were 69.91 and 64.69% respectively which were diminished >20%. Domestic processing reduced H_2_O_2_, hydroxyl, superoxide, DPPH and ABTS radical scavenging activity in performed in vitro assays which were further supported by in vivo CAT and SOD assay. Heating along with use of excess water make the hydrophilic flavonols and flavonoids prone to lose. Nevertheless, antioxidant activities were high enough to normalize the oxidative damage.

In LPO assay, the unprocessed flour showed comparatively lower inhibition activity than the processed flour and this activity was also concentration dependent. At 400 mg/kg dose, LPO inhibition activity were 60.71 and 15.75% in liver and serum respectively which were boosted up to 65.24 and 22.42% in turn. In vitro LPO inhibition and anti-hemolytic activity were also increased after processing. These may support the logics behind excellent LPO inhibition potential of lupin cultivars. Protein is one of the vital targets for all types of oxidants. Protein content in liver sample was determined in this investigation to examine antioxidant defense against protein degradation by ROS and free radicals. Protein content in liver sample of lupin flour treated mice was satisfactory (260.76 mg/mL) compared to standard (334.89 mg/mL). After processing, it was decreased to 111.53 mg/mL.

Flavonoids are solely responsible for hepatoprotective action. Catechin, apigenin, rutin, quercetin, venoruton and maringenin have proven strong hepatoprotective activities in previous study [80]. Another study showed hepatoprotective effects of flavonoid at 1–100 μg/mL concentration by reducing aspartate aminotransferase (AST) and alanine aminotransferase (ALT) leakage from hepatocytes in CCl_4_ induced oxidative mice. As for serum albumin, hydroxyproline and sialic acid in liver were also reduced by flavonoid treatment [81]. Oxidants in liver homogenate were reduced significantly after administration of lupin flours in this study. The outcomes of these in vivo studies suggest that lupin has hepatoprotective activities because of its rich flavonol and flavonoid contents.

## 4. Conclusions

All nine lupin cultivars tested exhibited varieties of bioactive compounds with considerable antioxidant activity. The major compounds found in all cultivars were hexadecanoic acid methyl ester, 9,12-octadecadienoic acid methyl ester, methyl stearate, lupanine,13-docosenamide and 11-octadecenoic acid (Z)- methyl ester. Many of the compounds declined and disappeared after processing, whereas several compounds were increased remarkably, such as lupanine and 13-docosenamide. The key reasons behind increasing these two bioactive chemicals were thermal release of unbound secondary metabolites and their non-volatile nature. Both have high boiling points and are stable at high temperatures. Processing at 100 °C may lead to concentrating these compounds.

A series of experimentations for assessing antioxidant activity of the nine cultivars of lupin were performed in terms of in-vitro and in-vivo techniques to safeguard actual-estimation, as well as diminish bias and over-substantiation. From the findings of these assays, it can be concluded that lupin seed is a high value source of antioxidant phytochemicals. Among the studied nine cultivars, Mandelup showed most potent antioxidant activity in both in vitro and in vivo assays. In addition, Jurien, Gunyidi and Barlock exhibited potential antioxidant activities. Lupin cultivars, are not only strong scavengers of ROS and free radicals, but also good stabilizers of oxidative-prone substances. They have great potentiality to accelerate CAT and SOD activities, either by stimulating their functionality or eliminating free radicals and ROS or both. Hepato-destructive oxidants were found to be reduced by lupin flours significantly which applaud their hepatoprotective effects. Moreover, protein degradation was slackened by administration of lupin flours.

Domestic processing has very minor negative effects on antioxidant activity of lupin flours in most of the assays except LPO and anti-hemolytic assays. LPO and anti-hemolytic activity were speeded up by processing. Total phenolics were augmented significantly due to heating, whereas flavonoids and flavonols were affected somewhat by excess water usage during processing rather than heat. A special instruction about domestic processing concerning short time heating at <100 °C and the use of minimal water can make sure maximum antioxidant activities in lupin flours.

The studied Australian lupin flours are easy to cultivate, process and palatable. Additionally they are cheap and rich in potential antioxidants, fiber and proteins. They can be an effective and valuable source of functional foods and nutraceuticals with special instructions about processing techniques to developing, as well as developed populations.

## Figures and Tables

**Figure 1 antioxidants-09-00282-f001:**
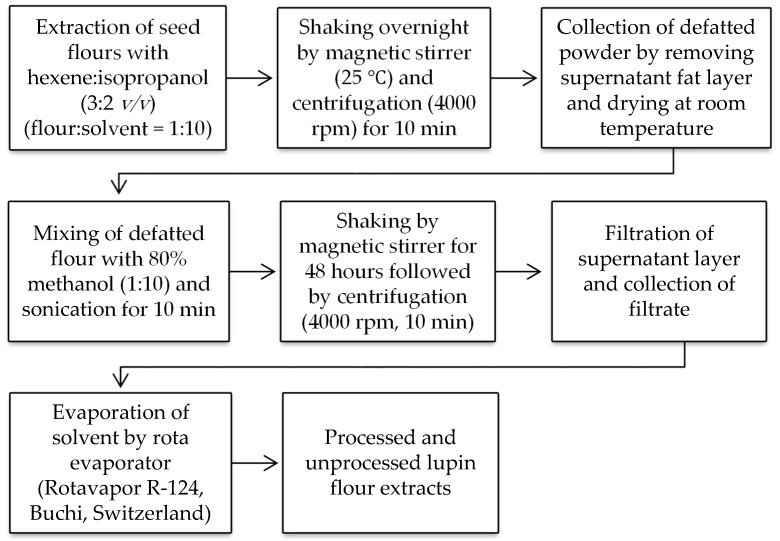
Schematic representation of preparation of unprocessed and processed lupin flour extract. Extraction process was carried out three times for each cultivar of both processed and unprocessed lupin flours.

**Figure 2 antioxidants-09-00282-f002:**
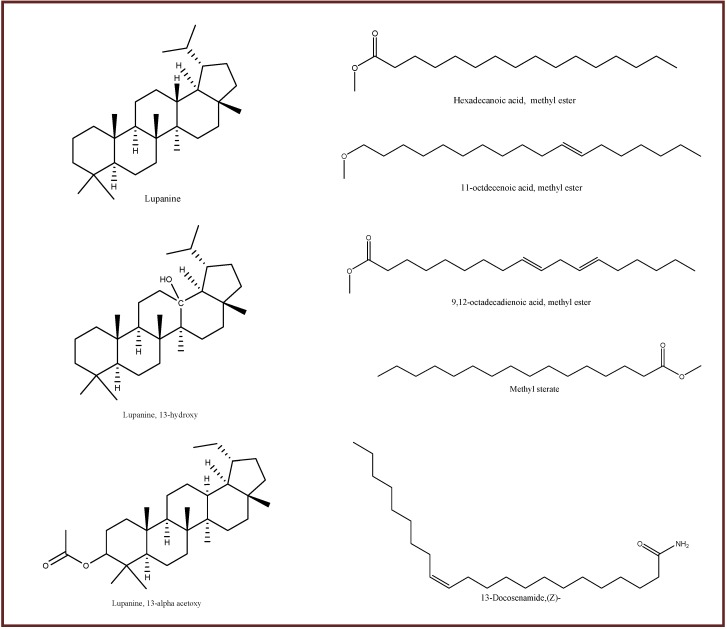
Major bioactive compounds present in nine cultivars of lupin species in both unprocessed and processed conditions.

**Figure 3 antioxidants-09-00282-f003:**
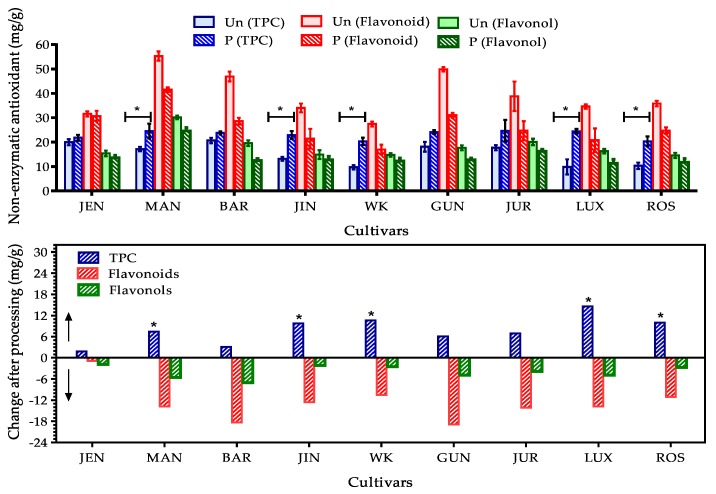
TPC, total flavonoids and flavonols contents of nine cultivars of lupin flour. The TPC, total flavonoids and flavonols contents of nine cultivars of lupin flour before and after processing. Values were represented as mean ± SEM where number of replicates was 3. ↔ Signified comparison of mean of processed vs. unprocessed flours. * Significance at p < 0.05 (unprocessed vs. processed condition). Upside arrow on top panel signified increase of TPC whereas downside arrow on bottom panel signified decrease of flavonoid and flavonol in processed flour compared to unprocessed one.

**Figure 4 antioxidants-09-00282-f004:**
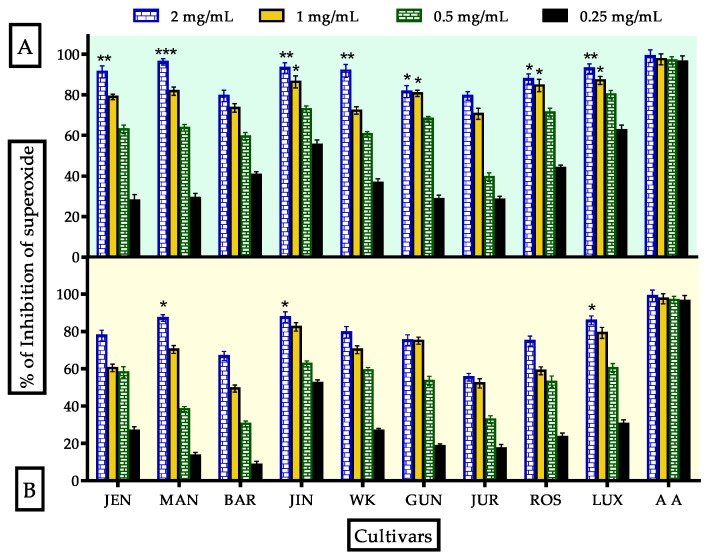
Antioxidant activity of unprocessed and processed lupin flours assayed by superoxide scavenging method; nine cultivars vs. ascorbic acid (standard). (**A**,**B**) indicated superoxide radical scavenging activity correspondingly. Data were stated as mean ± SEM where *n* = 3 (*** Significance at *p* < 0.001, ** significance at *p* < 0.01 and * significance at *p* < 0.05 vs. standard: ascorbic acid).

**Figure 5 antioxidants-09-00282-f005:**
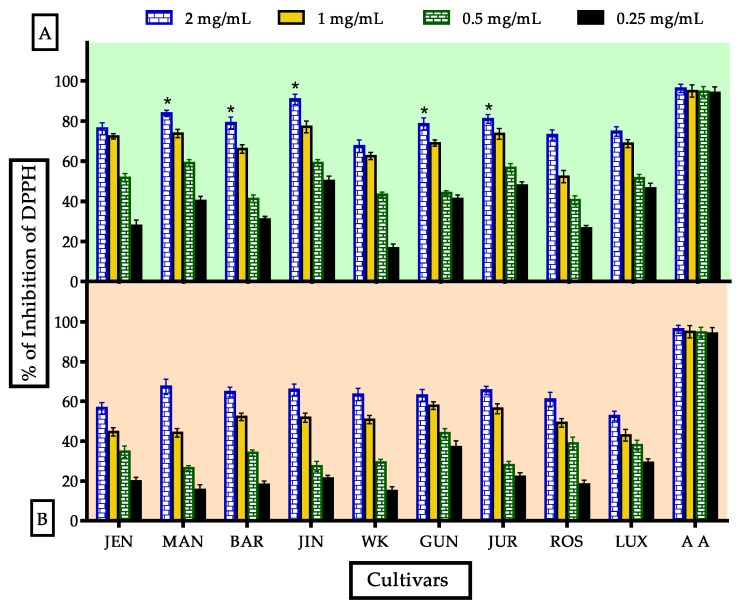
Antioxidant activity of unprocessed and processed lupin flours assayed by DPPH scavenging method; nine cultivars vs. ascorbic acid (standard). (**A**,**B**), indicated DPPH radical scavenging activity of unprocessed and processed flours respectively. Data were stated as mean ± SEM where *n* = 3 (* Significance at *p* < 0.05 vs. standard: ascorbic acid).

**Figure 6 antioxidants-09-00282-f006:**
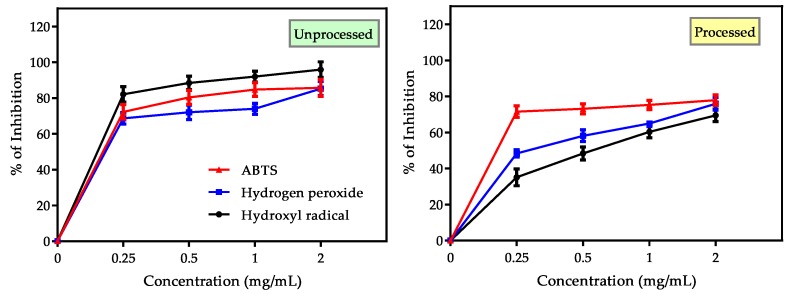
HAT based assay of Mandelup seed flours in unprocessed and processed conditions. Antioxidant potential of unprocessed and processed flours of most active cultivar: Mandelup assayed by HAT based (hydrogen peroxide, hydroxyl radical and ABTS radical scavenging) methods; each experiment was performed three times and data were expressed as mean ± SEM.

**Figure 7 antioxidants-09-00282-f007:**
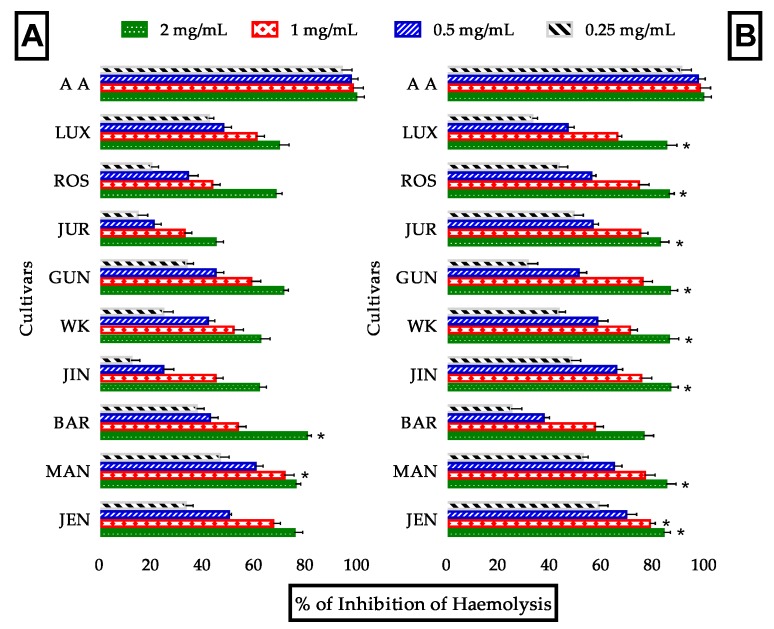
Antihaemolytic activity of lupin flours in unprocessed and processed conditions. Anti-haemolytic property of unprocessed (**A**) and processed (**B**) lupin flours; nine cultivars of lupin flours vs. ascorbic acid (standard). Data were presented as mean ± SEM of three replicates; (* Significance at *p* < 0.05 vs. standard: ascorbic acid).

**Figure 8 antioxidants-09-00282-f008:**
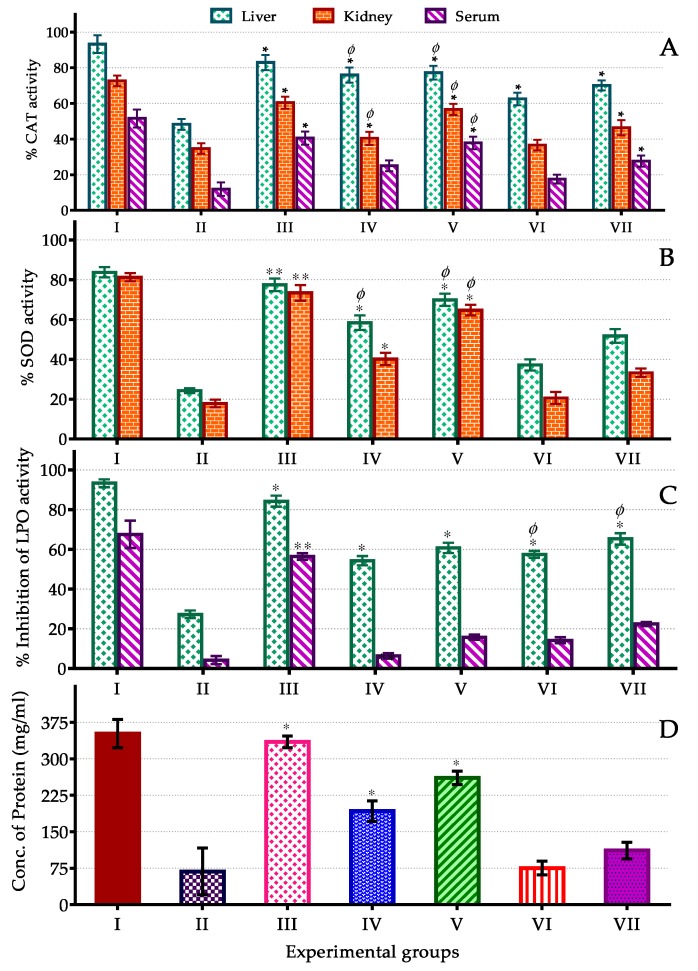
In-vivo antioxidant activities of unprocessed and processed lupin flours. The CAT, SOD, LPO activities and protein contents in serum, liver and kidney samples of CCl_4_ treated oxidative mice model after administration of processed and unprocessed flours of lupin cultivar Mandelup. Values were expressed as mean ± SEM where *n* = 6 mice in each groups and single and double stars indicated statistical significances (* *p* < 0.05; ** *p* < 0.01) vs. normal control group; *^Φ^ p* < 0.05 indicated significance vs. standard (by post-hoc Bonferroni test). (**A**–**D**) indicated activity of CAT (Catalase), SOD (Superoxide dismutase), LPO (Lipid peroxidation) and protein content, respectively. Group I: normal control, II: CCl_4_ control, III: standard (silymarin), IV: unprocessed flour (200 mg/kg), V: unprocessed flour (400 mg/kg), VI: processed flour (200 mg/kg) and VI: processed flour (400 mg/kg).

**Table 1 antioxidants-09-00282-t001:** Methods adopted for assessment of antioxidant activity of processed and unprocessed lupin flours.

Types of Method		Adopted Test Methods	Method Summary	Ref.
Assay of non-enzymatic antioxidant compounds		Total phenolic content	Folin Ciocalteu method	[15]
		Total flavonoid content	Colorimetric assay based on intensity of colour change of the mixture	[16,17]
		Total flavonol content		
In-vitro assay	Electron Transfer (ET) based assay	Superoxide radical scavenging assay	Measurement of reducing capacity of antioxidant (αH) in sample based on redox reaction (indicated by colour change) in presence of free radicals (R•) R• + αH + H_2_O → RH + α• + H_2_O	[3,18,19,20]
		DPPH• scavenging assay		
		TBARS assay		
	Hydrogen Atom Transfer (HAT) based assay	Hydroxyl radical scavenging assay	Quantitation of hydrogen atom donating capacity of antioxidant present in sample based on proton-coupled reaction (indicated by colour change) in presence of free radicals R• + αH → R^-^ + αH^+^•	[3,21,22,23]
		Hydrogen peroxide scavenging assay		
		ABTS^+^• scavenging assay		
	Others	Anti-hemolytic activity assay	Measurement of inhibiting capacity of RBCs hemolysis by ROS inducing oxidative stress	[24]
In-vivo assay		CAT activity assay	Quantifying the activity of enzymes responsible for suppressing ROS generation such as- catalase (CAT) and superoxide dismutase (SOD) in animal model treated with sample	[25,26]
		SOD activity assay		
		LPO activity assay	Measurement of the effect of sample on lipid peroxidation (auto-oxidation) and generation of malondialdehyde (MDA) in mice model	[19]
		Estimation of protein	Assessment of total protein in oxidative stressed animal model treated with sample containing antioxidant	[27]

**Table 2 antioxidants-09-00282-t002:** Major bioactive compounds present in all of nine cultivars of unprocessed and processed lupin seed flours.

Cultivars		Bioactive Compounds Concentration (%)					
		A	B	C	D	E	F
BAR	Un	31.06 ± 6.29	4.64 ± 2.10	1.07 ± 0.09	3.95 ± 2.11	1.83 ± 1.01	20.04 ± 8.07
	P	5.73 ± 3.38	2.22 ± 1.07	0.33 ± 0.13	4.39 ± 1.92	3.58 ± 2.31	48.75 ± 19.04
GUN	Un	8.52 ± 2.11	4.01 ± 2.22	0.95 ± 0.77	1.12 ± 1.02	1.47 ± 0.91	16.87 ± 3.92
	P	6.28 ± 2.09	1.45 ± 0.77	0.35 ± 0.13	1.42 ± 1.11	1.91 ± 0.22	21.73 ± 6.22
JEN	Un	25.92 ± 7.20	7.52 ± 3.10	0.78 ± 0.27	2.93 ± 2.00	2.73 ± 0.92	31.74 ± 6.31
	P	2.87 ± 0.91	1.58 ± 1.07	0.35 ± 0.03	4.09 ± 3.81	3.30 ± 1.00	37.42 ± 9.28
JIN	Un	2.29 ± 1.88	0.59 ± 0.23	1.21 ± 1.01	3.39 ± 0.87	3.68 ± 2.13	13.60 ± 4.08
	P	1.32 ± 1.01	0.35 ± 0.13	0.65 ± 0.92	4.29 ± 0.02	4.70 ± 2.01	12.84 ± 4.98
JUR	Un	2.91 ± 1.88	2.09 ± 1.11	2.31 ± 0.21	1.44 ± 0.22	3.59 ± 3.00	18.82 ± 7.30
	P	20.70 ± 2.03	3.91 ± 2.08	0.25 ± 0.03	2.79 ± 1.99	4.91 ± 1.98	26.44 ± 3.99
MAN	Un	18.30 ± 5.31	11.73 ± 3.40	0.84 ± 0.65	3.97 ± 1.00	0.72 ± 0.02	23.21 ± 6.31
	P	2.89 ± 0.98	4.82 ± 2.19	1.22 ± 1.13	0.73 ± 0.55	3.28 ± 2.03	27.98 ± 13.21
LUX	Un	18.22 ± 3.99	0.23 ± 0.16	0.74 ± 1.31	10.47 ± 3.21	0.24 ± 0.11	13.18 ± 2.99
	P	2.67 ± 2.02	0.59 ± 0.03	1.76 ± 1.33	1.28 ± 1.10	7.08 ± 0.63	16.43 ± 1.31
ROS	Un	0.23 ± 0.41	2.12 ± 2.00	2.13 ± 2.09	5.22 ± 2.03	2.30 ± 1.00	1.66 ± 0.51
	P	0.11 ± 0.39	0.22 ± 0.17	0.94 ± 0.88	2.39 ± 0.81	5.12 ± 2.62	3.22 ± 1.07
WK	Un	10.08 ± 1.49	3.96 ± 2.18	0.54 ± 0.41	8.91 ± 2.22	6.36 ± 3.19	18.33 ± 3.88
	P	3.29 ± 1.00	1.24 ± 0.91	3.88 ± 2.14	0.53 ± 0.44	8.75 ± 5.66	38.25 ± 14.23

A, B, C, D, E and F indicated hexadecanoic acid, methyl ester; 9,12-octadecadienoic acid, methyl ester; 11-octadecenoic acid, methyl ester; methyl stearate; lupanine and 13-docosenamide, (Z)- respectively. Data was represented as mean ± SEM.

**Table 3 antioxidants-09-00282-t003:** IC_50_ values of highly active cultivars in different types of in-vitro antioxidant test.

IC_50_ Values (mg/mL)		Cultivars/Compounds					
		MAN	JUR	BAR	GUN	A A	Trolox
DPPH	Un	193.8 ± 7.8	676.6 ± 16.1	591.9 ± 21.9	660.9 ± 70.1	56.6 ± 2.33	--
	P	1898 ± 72.4	751.3 ± 25.3	746.5 ± 18.0	811 ± 22.0		
ABTS	Un	366 ± 10.3	--	156.2 ± 7.3	349.6 ± 8.2	--	137 ± 4.28
	P	511.2 ± 22.2	--	223.2 ± 10.2	455.6 ± 14.4		
LPO	Un	487.7 ± 34.2	323.9 ± 8.4	530.8 ± 19.0	584 ± 13.8	91.3 ± 3.89	--
	P	369.8 ± 9.3	157.7 ± 7.6	368 ± 12.9	345.3 ± 7.3		
Hydroxyl	Un	553.2 ± 29.2	723.3 ± 14.9	1040 ± 47.5	519.5 ± 16.4	126.2 ± 4.22	--
	P	673.1 ± 18.2	819 ± 17.2	1198 ± 90.9	553.2 ± 31.5		
Anti-hemolytic	Un	426.2 ± 12.2	475.5 ± 13.3	580.2 ± 31.2	572.1 ± 19.7	237.1 ± 2.39	--
	P	500 ± 20.0	716.7 ± 11.8	937 ± 35.9	802.2 ± 41.0		
Superoxide	Un	552.2 ± 28.2	535.8 ± 19.3	376.2 ± 12.2	366.2 ± 11.0	276 ± 3.37	--
	P	604.9 ± 13.3	679.8 ± 16.1	759 ± 18.3	465.1 ± 10.6		
H_2_O_2_	Un	376.7 ± 8.2	340 ± 31.0	521.8 ± 27.7	532.4 ± 23.8	23.8 ± 1.97	--
	P	690 ± 12.7	561 ± 7.9	677.4 ± 22.9	883.5 ± 32.3		

Data was presented as mean ± SD, where *n* = 3. Un and P indicated unprocessed and processed flours respectively, (--) denoted not done. MAN, JUR, BAR and GUN were highly potent cultivars and A A (ascorbic acid) and trolox were standard used in assay.

**Table 4 antioxidants-09-00282-t004:** Activity of lupin flour (processed and unprocessed) in LPO activity assayed by TBARS method using egg-yolk and bovine brain homogenate.

Cultivars	TBARS Assay (Egg-Yolk and Bovine Brain Homogenate)							
	2 mg/mL		1 mg/mL		0.5 mg/mL		0.25 mg/mL	
	Un	P	Un	P	Un	P	Un	P
JEN ^1^	76.12 ± 2.49	93.66 ± 4.01 ^b^	62.62 ± 1.34	91.58 ± 3.02 ^bd^	58.46 ± 2.11	90.18 ± 1.9 ^bd^	47.76 ± 2.93	87.17 ± 3.19 ^ac^
MAN ^1^	70.82 ± 3.16	93.25 ± 3.98 ^bc^	69.26 ± 2.02	87.22 ± 4.56 ^a^	45.06 ± 2.62	84.47± 3.67 ^ac^	17.96 ± 2.22	59.29 ± 2.09 ^d^
MAN ^2^	82.97 ± 2.99	88.01 ± 4.12 ^a^	76.85 ± 2.76	83.57 ± 3.76	72.78 ± 4.01	80.69 ± 2.69	70.50 ± 2.67	75.06 ± 3.43
BAR ^1^	56.65 ± 2.17	88.31 ± 2.88 ^ad^	44.54 ± 2.09	73.05 ± 2.98	37.90 ± 3.92	67.65 ± 2.93	16.77 ± 1.56	43.45 ± 4.12
JIN ^1^	61.01 ± 2.26	95.53 ± 3.06 ^bd^	42.93 ± 2.96	92.52 ± 2.88 ^bd^	40.39 ± 2.61	87.85 ± 2.64 ^c^	26.01 ± 2.34	68.48 ± 4.67
JIN ^2^	87.65 ± 3.12 ^a^	88.37 ± 3.09 ^a^	84.77 ± 1.97 ^a^	86.45 ± 2.67 ^a^	78.65 ± 3.96	84.4 ± 3.79 ^a^	62.58 ± 2.83	66.43 ± 3.93
WK ^1^	88.73 ± 3.21 ^a^	90.60 ± 1.99 ^a^	70.19 ± 3.81	84.42 ± 3.71 ^a^	46.41 ± 1.23	70.87 ± 4.04	21.65 ± 3.99	51.55 ± 3.97
GUN ^1^	87.64 ± 2.38 ^a^	94.39 ± 2.78 ^b^	70.9 8 ± 3.52	86.55 ± 1.97 ^a^	54.10 ± 1.12	65.93 ± 3.98	32.29 ± 2.01	50.67 ± 2.78
JUR ^1^	85.72 ± 2.92	95.01 ± 4.28 ^b^	81.88 ± 2.73	87.79 ± 1.29 ^a^	72.89 ± 2.09	83.59 ± 3.42	11.37 ± 3.73	77.62 ± 3.29 ^c^
JUR ^2^	88.24 ± 3.34 ^a^	89.33 ± 2.67 ^a^	84.05 ± 2.85 ^a^	87.53 ± 3.43 ^a^	82.97 ± 3.19	85.97± 1.88 ^a^	79.61 ± 3.09	84.53 ± 4.13 ^a^
LUX ^1^	86.34 ± 2.71	95.11 ± 2.44 ^b^	51.77 ± 3.02	85.61 ± 3.43 ^ac^	40.18 ± 2.13	72.48 ± 2.98	23.52 ± 1.46	57.89 ± 1.09
ROS ^1^	76.64 ± 2.92	95.84 ± 3.78 ^b^	69.26 ± 4.33	92.10 ± 3.91 ^c^	52.44 ± 2.01	88.52 ±1.87 ^ac^	50.13 ± 2.60	70.82 ± 2.09
A A ^1^	92.89 ± 3.04	92.88 ± 3.04	91.92 ± 2.67	91.92 ± 2.67	90.52 ± 3.11	90.52 ± 3.11	89.23 ± 2.87	89.23 ± 2.87
A A ^2^	92.88 ± 3.04	92.89 ± 3.04	91.89 ± 2.67	91.85 ± 2.67	90.52 ± 3.11	90.52 ± 3.11	89.23 ± 2.87	89.23 ± 2.87

^1, 2^ indicated egg-yolk homogenate and bovine brain homogenate respectively. ^a^ and ^b^ signified different level of significance (^a^ indicated significance at *p* < 0.05 and ^b^ indicated significance at *p* < 0.01 vs. standard: ascorbic acid). ^C^ (*p* < 0.05) and ^d^ (*p* < 0.01) represented significance vs. unprocessed flour (Student’s t test). Values were represented as mean ± SEM, where *n* = 3.

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
