# Peer review of "Bioactive Variability and In Vitro and In Vivo Antioxidant Activity of Unprocessed and Processed Flour of Nine Cultivars of Australian lupin Species: A Comprehensive Substantiation"

_antioxidants, 2020, doi:10.3390/antiox9040282_

Round 1

Reviewer 1 Report

This manuscript written by Kishor Mazumder and coworkers described the antioxidant activity of unprocessed and processed flour of nine cultivars of Australian lupin species. The authors spent a lot of efforts to prepare their samples and assess the antioxidant activity by in-vitro and in-vivo techniques.

I think the data are sound and this manuscript can be published in Antioxidants after minor revision.

The comments are list below:

Line 272, “3”. Results and Discussion For figure 2 and figure S1. Please redraw the structures using “ACS document1996” format. Figure 2, the structure of Lupanine, please avoid protons H-13/H-18 and H-5/methyl touch each other. The structure of Lupanine, 13-hydroxy is wrong. Line 583, the authors mentioned two compounds increased after processing. I hope the authors can explain the reasons and add some sentences in the manuscript.

Author Response

Comment-1

Line 272, “3”. Results and Discussion For figure 2 and figure S1. Please redraw the structures using “ACS document1996” format.

Response: All structures are drawn by using ACS documents 1996 format.

Comment-2

Figure 2, the structure of Lupanine, please avoid protons H-13/H-18 and H-5/methyl touch each other. The structure of Lupanine, 13-hydroxy is wrong.

Response: Structure of Lupanine and Lupanine, 13-hydroxy are refurnished as per comment.

Comment-3

 Line 583, the authors mentioned two compounds increased after processing. I hope the authors can explain the reasons and add some sentences in the manuscript.

Response: Reason for increasing two vital compounds have been explained and substantiated. Please see line 640-643.

Reviewer 2 Report

The manuscript by Mazumder M et al., is a detailed study on the bioactive compounds and the antioxidant activities of nine cultivars of Australian lupin species and comparison of their antioxidant activities under domestic heat processing using in vitro an in vivo approaches. The study has interesting findings that would be of interest to food science audience. Please find below some comments that may help improve the overall presentation. 

Lines 28-30 - Please rephrase.  Figure 3. Please indicate the statistical significance between groups in the legend.  Table 2. There is no information on the number of replicates of the assays summarised in the table. IC50 values should be presented as MEAN ± SD.  line 580: better use the word "antioxidant" instead of "pharmacological" Avoid overstatements in the manuscript such as line 32 "to get rid of oxidative stress induced diseases and disorders", lines 606-606 "..used in oxidative damage related pathological conditions as well as maintenance of sound and vigorous physiology"

Author Response

Comment-1: Lines 28-30 - Please rephrase. 

Response: Lines 28-30 are rephrased.

Comment-2: Figure 3. Please indicate the statistical significance between groups in the legend.

Response: Done as per comment. Please see figure 3.

Comment-3: Table 2. There is no information on the number of replicates of the assays summarized in the table. IC50 values should be presented as MEAN ± SD.

Response:  Number of replicate is specified in Table 2. IC50 values are represented as mean ± SD.

Comment-4: line 580: better use the word "antioxidant" instead of "pharmacological"

 Response: Word "antioxidant" is  used instead of "pharmacological" at that line (now changed to line 636 due to correction).

Comment-5:  Avoid overstatements in the manuscript such as line 32 "to get rid of oxidative stress induced diseases and disorders", lines 606-606 "..used in oxidative damage related pathological conditions as well as maintenance of sound and vigorous physiology"

Response: Overstatements are removed.

Reviewer 3 Report

The authors presented a multidirectional study that is consistent with the aim of the journal.

Most of comments are referred to the employed animal experimental model.

How did authors calculate the total animal number and above all the animal number per group?

Was it statistically significant?

The CCl4 toxicity model should be substantiated by appropriate reference in the related descriptive paragraph.

Furthermore, how did the authors decide the extract dosage to administer to mice?

Although authors indicated ip route for CCl4 administration, it is not clear in the text if the same route was used for administering sylimarin and the extracts. Please clarify.

Finally, the authors did not report the sacrifice method. It is mandatory to include in the material and methods.

The statistical approach is good. On the other hand, with the aim to improve the description of significant data, I suggest to include the ANOVA P values in the figure captions, alongside with the P values related to post hoc test.

Overall, I suggest the publication of the manuscript after revision.

Author Response

Comment-1: How did authors calculate the total animal number and above all the animal number per group?

Response: Total animal number and the number of animal per group were calculated by resource equation considering degree of freedom of ANOVA (excluding control group as mice of this group didn’t receive CCl4 and any chemicals). Then the number of animal was corrected by considering 10% attrition of sample size.

Comment-2: Was it statistically significant?

Response: As it was calculated by considering statistical approach, it is statistically significant.

Comment-3: The CCl4 toxicity model should be substantiated by appropriate reference in the related descriptive paragraph.

Response: CCl4 toxicity method has been substantiated by appropriate reference. Please see ref: 28 (line 754) in reference section. Detailed mechanism underlying oxidative damage caused by CCl4 was described in result and discussion, section 3.4, line 581-587.

Comment-4: Furthermore, how did the authors decide the extract dosage to administer to mice?

Response: Extract dosage was calculated after performing acute toxicity study. There was no sign of death or abnormalities at a dose of 2000 mg/kg (IP route). The toxic dose was divided by 10 and 20 and the obtained dosage was taken to study.

Comment-5: I suggest to include the ANOVA P values in the figure captions, alongside with the P values related to post hoc test.

Response: ANOVA P values are added along with P values related to post hoc test.

Comment-5: Although authors indicated ip route for CCl4 administration, it is not clear in the text if the same route was used for administering sylimarin and the extracts. Please clarify.

Response: The route for administering the sylimarin and the extracts have been clarified. Please see section 2.7.2, line 243.

Comment-6: Finally, the authors did not report the sacrifice method. It is mandatory to include in the material and methods.

Response: The sacrifice method has been included. Please see section 2.7.2, lines 245.

Reviewer 4 Report

In the paper Authors discuss the results of an in-depth study on the in vitro and in vivo antioxidant capacity of lupin flours coming from nine cultivars before and after cooking.

Results are worthy to be published. I invite Authors to re-write the paper taking into account the comments underneath and resubmit it, as the manuscript needs extensive major revisions before to be accepted for publication.

Comments:

Materials and Method section:

In this section there are various criticisms, as follows:

Starting from line 99, the numbering of the subsections is wrong and does not correctly follow the progression of numbers both for the main subsections (2.N) and the sub-sub-sections (2.N.n). Please correct Materials lines 96-97: please specify here between brackets the abbreviation of the name of each cultivar which are then used in text, tables and figures. Preparation of samples: Figure 1 mainly regards the preparation of extracts from uncooked and cooked lupin seed flours; the preparation of flours is at the first two steps of the flow-sheet. A suggestion is to describe in the text under the materials sub-section the content of these two steps, adding further technological details (such as the equipment used for dehulling, cooking and drying, the water/flour ratio used in the cooking process and the time of cooking), delete the first two steps from the flow-sheet and modify the tittle of the figure as actually it is the flow sheet of the extraction procedure followed to obtain extracts for all the tests and analyses carried out in the research. For flour preparation (uncooked and cooked), please specify the number of replicates per cultivar. GC-MS analysis: please specify at the end of the sub-section how did you express GC-MS data. Furthermore, please give details on other methods used to confirm the GC-MS identification, such as retention index consistent with data from the literature for the same phase, comparison of retention data with those of authentic standards. Table 1 at lines 127-130 does not summary the methods listed at lines 121-125, but reports the percentages of the main bioactive compounds for uncooked and cooked flours. The summary of methods is in Table S1, hence replace at line 126 Table 1 with table S1. Line 137: please give details (type and model) of the UV-visible spectrophotometer used. Line 166: please re-phrase as it is a little bit confusing Line 249: malondialdehyde instead of MDA.

Results and Discussion:

GC-MS analysis: please re-write all the subsection referring also to the proper supplementary materials and moving under this sub-section the Table 1 of lines 127-130. I suggest to re-organize this sub-section:

Firstly, describe the example of TIC from GC-MS analysis of figure S2 (Please in both panels of Figure S2 replace the peak number with the name of the compound identified) Secondly, give a description of the table of identified compounds (for example the number of compounds per class of compounds) listed in Table S2 and then describe the differences in bioactive profile among cultivars and with cooking (data of Tables S3a and S3b) In the end, a more in-depth description of the six main compounds of Table 1 Table S2: in order to be more informative on GC-MS results additional columns for retention index and ion fragments with relative percentage amounts have to be inserted between the first column (bioactive compounds) and pharmacological activity) I suggest to merge Tables S3a and S3b into a table with the horizontal layout of the page in order to have data of all the cultivars in the same row. To improve readability, group the compounds according to their category, and uniform through all the table the number of decimal digits (a suggestion 2 decimal digits are enough). Please specify if these data are mean values of the replicates of the processing. Table 1: please use 2 decimal digits for all the percentage numbers, add standard error and the results of >Student’s t test for the comparison between Un and P flour for each cultivar.

Figure 3: It is not specified in the figure caption the meanings of one star plus bar above TPC data of the top panel and the star at the top of TPC change of the bottom panel. Please add explanation.

Table 2 line 411: please adjust text in order to have the whole table in a single page. Add the standard error.

Figures 4: I suggest to split this figure in two (each one dedicated to one test) in order to improve the layout of bar graphs by enlarging the pitch of the characters and symbols

Table 3 line 495: please explain for cultivar without upper-case number, if data are the average of the two homogenates. Please specify the level of significance for a and b in the footnote. In order to say that there is an increase of inhibition with cooking as stated at lines 480-481 the Student’s t test has to be carried out. I suggest to split the comments of the comparison with the performance of AA from the comparison between uncooked and cooked flours.

Figure 7: please enlarge the pitch of all characters as it is very difficult to read. Line 551. Probably the second unprocessed refer to processed (cooked) please correct.

References: revise carefully all references and make suitable correction to fit the Instruction for Authors, i.e.: Author 1, A.B.; Author 2, C.D. Title of the article. Abbreviated Journal Name YearVolume, page range.

Line 625: replace “9” with “9(8), 14-32.”

Line 641: replace “v 25, pp. 190-197-2012 v.2025 no.2012” with “25(2),190-197.”

Line 748: replace “1” with “1, 1106387”

Line 753: delete “#x103; tra,”

Line 755: replace “2016, 8” with “2018, article ID 7638542, 8 pages”

Line 791: replace “$V 10” with “10(3)”

Author Response

Comments:

Materials and Method section:

Comment-01: Starting from line 99, the numbering of the subsections is wrong and does not correctly follow the progression of numbers both for the main subsections (2.N) and the sub-sub-sections (2.N.n). Please correct.

Response: Numbers for main subsections and sub-sub-sections are corrected from line 93 (sub-section 2.2) to results and discussion section.

Comment-02: Materials lines 96-97: please specify here between brackets the abbreviation of the name of each cultivar which are then used in text, tables and figures.

Response: Abbreviations of the name of each cultivar are specified by brackets in sub-section 2.2 (lines 94-96). Abbreviated names are the added in text, tables and figures.

Comment-03: Preparation of samples: Figure 1 mainly regards the preparation of extracts from uncooked and cooked lupin seed flours; the preparation of flours is at the first two steps of the flow-sheet. A suggestion is to describe in the text under the materials sub-section the content of these two steps, adding further technological details (such as the equipment used for dehulling, cooking and drying, the water/flour ratio used in the cooking process and the time of cooking), delete the first two steps from the flow-sheet and modify the tittle of the figure as actually it is the flow sheet of the extraction procedure followed to obtain extracts for all the tests and analyses carried out in the research.

Response: Sample preparation is described and reorganized according to suggestion. First two steps are described as preparation of flours (subsection 2.2.1) and then, flow chart is rearranged under subsection 2.2.2 as extraction process adding technological details of equipment used and the cooking conditions and time. Please see section 2.2.1 and 2.2.2 line 98-112.

Comment-04: For flour preparation (uncooked and cooked), please specify the number of replicates per cultivar.

Response: Replicate number for flour preparation is specified in section 2.2.1 (line 101) and section 2.2.2 (line 111).

 Comment-5: GC-MS analysis: please specify at the end of the sub-section how did you express GC-MS data. Furthermore, please give details on other methods used to confirm the GC-MS identification, such as retention index consistent with data from the literature for the same phase, comparison of retention data with those of authentic standards.

Response: GC-MS data expression method is specified in section 2.3, line 124.

Comment-6: Table 1 at lines 127-130 does not summary the methods listed at lines 121-125, but reports the percentages of the main bioactive compounds for uncooked and cooked flours. The summary of methods is in Table S1, hence replace at line 126 Table 1 with table S1.

Response: Table 1 is replaced with table S1. Please see section 2.4, line 135.

Comment-7: Line 137: please give details (type and model) of the UV-visible spectrophotometer used.

Response: Type and model of UV-visible spectrophotometer used in this research work is added in line 144.

Comment-8: Line 166: please re-phrase as it is a little bit confusing

Response: The line is rephrased. Please see section 2.6.2., line 174.

Comment-9: Line 249: malondialdehyde instead of MDA.

Response: Malondialdehyde is added as per comment. Please see section 2.7.5, line 257.

Results and Discussion:

Comment-10: GC-MS analysis: please re-write all the subsection referring also to the proper supplementary materials and moving under this sub-section the Table 1 of lines 127-130. I suggest to re-organize this sub-section:

Firstly, describe the example of TIC from GC-MS analysis of figure S2 (Please in both panels of Figure S2 replace the peak number with the name of the compound identified)

Secondly, give a description of the table of identified compounds (for example the number of compounds per class of compounds) listed in Table S2 and then describe the differences in bioactive profile among cultivars and with cooking (data of Tables S3a and S3b) In the end, a more in-depth description of the six main compounds of Table 1

Table S2: in order to be more informative on GC-MS results additional columns for retention index and ion fragments with relative percentage amounts have to be inserted between the first column (bioactive compounds) and pharmacological activity)

I suggest to merge Tables S3a and S3b into a table with the horizontal layout of the page in order to have data of all the cultivars in the same row.

To improve readability, group the compounds according to their category, and uniform through all the table the number of decimal digits (a suggestion 2 decimal digits are enough). Please specify if these data are mean values of the replicates of the processing. Table 1: please use 2 decimal digits for all the percentage numbers, add standard error and the results of >Student’s t test for the comparison between Un and P flour for each cultivar.

Response: Result and discussion of GC-MS section is rewritten according to instructions. Table 2 (table number is changed from 1) is moved under this section.

Name of major compounds are added in figure S2. Due to lack of space between peaks, all the compounds identified are not added.

Table S1 is reorganized by using class of compounds, retention time and m/z data.

Table S2 is reordered horizontal layout by combining table S3a and S3b.

Student’s t test was performed for table 2. No significance was observed.

Comment-11: Figure 3: It is not specified in the figure caption the meanings of one star plus bar above TPC data of the top panel and the star at the top of TPC change of the bottom panel. Please add explanation.

Response: Figure caption for figure 3 is corrected as per comment.

Comment-12: Table 2 line 411: please adjust text in order to have the whole table in a single page. Add the standard error.

Response: All the text, tables and figures are rearranged. Standard deviation is added with mean in table 2. Please check section 3.1, line 350.

Comment-13: Figures 4: I suggest to split this figure in two (each one dedicated to one test) in order to improve the layout of bar graphs by enlarging the pitch of the characters and symbols

Response: figure 4 is divided into two distinct figures (figure 4 and 5) and moving under each tests.

Comment-14: Table 3 line 495: please explain for cultivar without upper-case number, if data are the average of the two homogenates. Please specify the level of significance for a and b in the footnote. In order to say that there is an increase of inhibition with cooking as stated at lines 480-481 the Student’s t test has to be carried out. I suggest to split the comments of the comparison with the performance of AA from the comparison between uncooked and cooked flours.

Response: Table 3 (now table 4) upper case numbers are corrected. Results of student’s t test are inserted also. The comment is divided into two parts as per instruction. Please see section 3.3.4, table 4.

Comment-15: Figure 7: please enlarge the pitch of all characters as it is very difficult to read. Line 551. Probably the second unprocessed refer to processed (cooked) please correct.

Response: In figure 7 (now fig 8), pitch of all characters are enlarged. Second unprocessed is replaced with processed at line 551 (now line 594)

Comment-16: References: revise carefully all references and make suitable correction to fit the Instruction for Authors, i.e.: Author 1, A.B.; Author 2, C.D. Title of the article. Abbreviated Journal Name YearVolume, page range.

Line 625: replace “9” with “9(8), 14-32.”

Line 641: replace “v 25, pp. 190-197-2012 v.2025 no.2012” with “25(2),190-197.”

Line 748: replace “1” with “1, 1106387”

Line 753: delete “#x103; tra,”

Line 755: replace “2016, 8” with “2018, article ID 7638542, 8 pages”

Line 791: replace “$V 10” with “10(3)”

Response: References were corrected according to the instructions-

Line 625 (now line 684): “9” is replaced with “9(8), 14-32.”

Line 641 (now line 700): “v 25, pp. 190-197-2012 v.2025 no.2012” is replaced with “25(2),190-197.”

Line 748 (now line 788): “1” is replaced with “1, 1106387”

Line 753 (now line 820): “#x103; tra,” is deleted.

Line 755 (now line 822): “2016, 8” is replaced with “2018, article ID 7638542, 8 pages”

Line 791 (now line 860): “$V 10” is replaced with “10(3)”

Round 2

Reviewer 4 Report

The revised manuscript can be accepted for publication in the present form.

Only few additional comments:

Line 273: delete “Oliver H.”

Line 365: “are shown” instead of “were shown”

Line 418: “Yu et al. [55]” instead of “Yu L et al [56]”

Line 426: please add “[56]” the number of reference Chaaban et al.

As for reference list at lines 684-922, please revise carefully all references and make suitable correction to fit the Instruction for:

- Authors names: reference 48: please write authors’ names not all in capital letters

- the title of the article: reference 3: please write title not all in capital letters

- the journal name: please check all reference and use only the abbreviated form for the journal name.

Round 2

  1. Comment: Line 273: delete “Oliver H.”

Response: in Line 273, "Oliver H." has been deleted.

  1. Comment: Line 365: “are shown” instead of “were shown”

Response: Line 365: “Were shown” has been replaced by “are shown”.

  1. Comment: Line 418: “Yu et al. [55]” instead of “Yu L et al [56]”

Response: Line 418: “Yu et al. [55]” is used instead of “Yu L et al. [56]”.

  1. Comment: Line 426: please add “[56]” the number of reference Chaaban et al.

Response: Line 429:Reference number for Chaaban et al. is added which is [56].

  1. Comment: As for reference list at lines 684-922, please revise carefully all references and make suitable correction to fit the Instruction for:

Response: Line 684-922: All the references are checked carefully and abbreviated forms of journal names are used. Others related issues are also edited as per instructions in journal referencing guidelines.

  1. Comment:- Authors names: reference 48: please write authors’ names not all in capital letters

Response: Reference 48: Authors’ names are corrected as per journal’s referencing instructions

  1. Comment: - the title of the article: reference 3: please write title not all in capital letters

Response: Reference 3: Title of the article is corrected.

  1. Comment: - the journal name: please check all reference and use only the abbreviated form for the journal name.

Comment: in all references, journal names are placed in abbreviated form.